# LGBTQ+ identities in the Indian audiovisual advertisements: A content analysis

**Khuman Bhagirath Jetubhai** 📧 *

School of Humanities and Social Sciences, Thapar Institute of Engineering & Technology, Patiala, Punjab, India

* bhagirathkhuman@yahoo.com

## Abstract

In the late 2000s, LGBTQ+ (lesbian, gay, bisexual, transgender, queer, and others) identities began to surface in Indian audiovisual marketing campaigns. By 2021, brands had produced at least 105 advertisements featuring LGBTQ+ identities. This research conducts an analysis of these campaigns, examining aspects such as their release date, featured characters, gender identity, sexual orientation, character age, narrative usage, genre, theme, and setting. The study reveals a substantial surge in advertising campaigns since 2018, with brands strategically launching them during Pride Month, Valentine's Day, and Women's Day. Characters embodying gay, lesbian, and transgender identities were most prevalent. Advertisements spotlighting gay, lesbian, or both gay and lesbian characters predominantly emphasised themes of love, featuring youthful characters and urban settings. In contrast, advertisements featuring transgender characters centred on human rights, with older characters and non-urban settings. Based on these findings, it is recommended that the number of LGBTQ+ marketing campaigns should continue to rise, diversifying character identities, ages, and settings while being released throughout the year.

**Data Availability Statement:** The data underlying the results presented in the study are available from: https://osf.io/56tmx.

**Funding:** The author(s) received no specific funding for this work.

## Introduction

Advertising serves as a powerful tool for capturing attention and promoting products [1]. Traditionally, advertising has played a significant role in gaining a profound understanding of both the present and potential audience for particular products, making advertisements an indispensable source of cultural insight [1]. Indeed, advertising has transcended its conventional role in marketing to emerge as a significant window into broader social developments [2]. Consequently, advertising and consumer identity share an intricate and inseparable connection. Piller [3] highlights that the reconfiguration of contemporary cultural identities is substantially influenced by the media, which plays a pivotal role in shaping these identities, characterised by their hybrid, intricate, and often paradoxical nature. In the context of queer identities, Tsai [4] underscores that the contemporary homosexual identity is a sociohistorical construct intricately intertwined with American media and market systems. Chasin further emphasises the role of advertising in nurturing a sense of community among sexual minority groups:

**Competing interests:** The authors have declared that no competing interests exist.

The national U.S. gay community came into being through the imagined comradeship of gay men and lesbians reading an increasingly commercial gay press. In that press, gay men and lesbians read for news of the growth of the movement, they read for news of consumption opportunities that reinforced their belonging in the community, and they read vernacular language that helped delineate the boundaries of the community [5].

In recent times, a distinctive identity-based segmentation strategy has facilitated the emergence of a niche market catering to LGBTQ+ people (for the purposes of this research, the LGBTQ+ acronym is used to encompass all gender identities and sexual orientations featured in Indian advertising campaigns). This strategy has significantly influenced the conceptualisation and societal understanding of gay identity within American consumer culture [4].

However, Bartholomew [6] observes both the merits and drawbacks of such niche advertisements. On the one hand, targeted advertising validates the identities of individuals who self-identify as gay, simultaneously promoting intragroup relationships as a means to accentuate distinctions from other societal groups. On the other hand, this marketing approach has given rise to a limited and narrowly defined understanding of what it means to be gay, how individuals can express their sexuality, and the restricted cultural spaces available for the development of a diverse gay identity. These limitations present a challenge to the process of self-categorization for individuals who identify as homosexually inclined [6].

Hence, the significance of advertisements arises from the intricate portrayal of LGBTQ + individuals within them, which holds considerable influence over both LGBTQ+ individuals themselves and the broader public's perception of this community. In India, a nation marked by an inconsistent legal journey towards LGBTQ+ rights—from 1994, when the initial petition challenging Section 377 was filed, to 2009, when the Delhi High Court ruled against it, then in 2013 when the Supreme Court reversed the High Court's decision, and ultimately in 2018 when the Supreme Court repealed Section 377, as documented by Thomas [7]—social acceptance of sexual and gender minority individuals is progressing slowly. In this context, marketing campaigns possess significant potential as a catalyst for societal transformation. This is primarily due to the profound impact of our interaction with and consumption of screen-based information on our perceptions of the world and ourselves [8].

To effectively implement campaigns centred on LGBTQ+ identities with the aim of driving transformative change among consumers, brands must establish a repository for tracking the evolution of advertising in this domain. Such a repository would enable the exploration of emerging insights into commercial advertising and its impact on the lives of LGBTQ+ citizens [1]. Furthermore, it would ensure accurate representation. Regrettably, while several repositories exist for Western advertisements, such as the Commercial Closet, Advertising Archives, and Ad Forum, none are available for Indian advertisements.

In addition to the absence of a repository, research on Indian LGBTQ+ advertisements is scarce. Khurana [9] conducted an examination of selected advertisements, aiming to discuss the marketing strategies that brands could employ to reach the relatively untapped LGBTQ + market in India, all while navigating the associated challenges and risks. Singh's work, referencing matrimonial print advertising in the Mid-Day newspaper and Myntra's "Bold is Beautiful" campaign, provided a perspective on how the Indian family had evolved into a pivotal "arbiter of queer relationality" [10]. Chauhan and Shukla [11] delved into the impact of "Bold is Beautiful" on viewers, concluding that there should be a concerted effort to create more advertisements with social messages to raise awareness of LGBTQ+ issues. Collectively, the research conducted within the Indian context pursued distinct objectives compared to the present study.

Given the scarcity of prior research in the Indian context, evaluating the strengths and weaknesses of Indian LGBTQ+ advertising, identifying areas for improvement, and drawing comparisons with Western counterparts present formidable challenges. This study endeavours to bridge this gap by meticulously documenting all LGBTQ+ marketing campaigns from their inception up to 2021 and conducting a comprehensive analysis of their content and context. The significance of this research becomes apparent within the context of the evolving LGBTQ + movement and the burgeoning interest of the market. Understanding how sexual and gender minority consumers are portrayed in the media takes on increasing importance [12]. Therefore, this paper seeks to address the following research questions:

1. To what extent has each passing year influenced the annual number of LGBTQ+-focused advertisements, and is this influence linked to specific events that occurred in those respective years?

2. In which month do we observe a significant surge in the number of advertisement releases, and does the presence of specific events in that month correlate with this increase in advertisements?

3. What gender and sexual identities are prominently featured or conspicuously absent in advertisements? Are transgender characters typically portrayed by transgender actors, or do cisgender actors commonly take on transgender roles in advertisements?

4. Is there a connection between the inclusion of themes related to love or human rights in advertisements and the presence of characters belonging to specific gender and sexual identities?

5. Does a character's age relate to their specific gender or sexual identity in advertisements?

6. Do the introductions of LGBTQ+ characters in advertisements often coincide with plot twists?

7. Is there a correlation between the urban or non-urban settings of advertisements and the specific gender or sexuality of the characters portrayed in them?

In essence, this study aims to gain insights into audio-visual advertisements that portray LGBTQ+ identities by examining the timing of campaign releases, the representation of LGBTQ+ characters, and the themes and settings within these advertisements.

## Methodology

To identify all Indian LGBTQ+ advertisement campaigns, I conducted a comprehensive search of web resources encompassing news and events related to brands and their marketing initiatives using popular search engines, including Google, Yahoo, and Bing. I refined the selection process by choosing web resources that had covered at least one LGBTQ+ marketing campaign. These campaigns were identified through the website search function of each resource. I examined campaigns released that were launched from their inception in 2008 until 2021., which corresponds to the timeframe of this research. The 16 selected web resources include: 1) exchange4media.com; 2) medianews4u.com; 3) mediasamosa.com; 4) campaignsoftheworld.com; 5) afaqs.com; 6) adgully.com; 7) brandinginasia.com; 8) campaign-india.in; 9) bestmediainfo.com; 10) campaignbriefasia.com; 11) theviralads.com; 12) clutter-cutters.in; 13) mediainfoline.com; 14) brandequity.economictimes.indiatimes.com; 15) mediabrief.com; 16) lgbtqindiaresource.in.

I established specific criteria to guide my selection of advertisements from the aforementioned web resources:

1. Advertisements, regardless of language and release year, were required to have been broadcast either on television or over the internet in India.

2. Advertisements had to be in audiovisual format since print advertisements did not allow for the comprehensive analysis of characters, plotlines, and settings.

As a result of this selection process, I amassed a total of 105 observations of marketing campaigns, which formed the basis for our content analysis. Content analysis is a "technique for making inferences by systematically and objectively identifying specific characteristics of messages" [13]. In this context, "messages" encompass a diverse range of forms, including spoken content such as speeches, letters, tweets, diaries, or blogs; written content like news reports; graphic content such as advertising illustrations and graffiti; and audio-visual content, including television shows, films, or computer games [14].

This study primarily focuses on conducting a content analysis of audiovisual advertisements. While data examination for content analysis can be approached using either quantitative or qualitative methodologies [15], this study employs quantitative tools to analyse the collected data. The subsequent subsection elaborates on how the coding process was carried out:

## 1. Advertisements released annually

To ascertain the number of advertisements released in a particular year and evaluate whether there was an increase, decrease, or no significant change in the number of advertisements released annually, I documented both the number of advertisements and their corresponding release years. Subsequently, I compared these figures to significant events that could have potentially influenced the number of advertisements released during each year.

## 2. Advertisements released monthly

To analyse the frequency of advertisements released in specific months and determine the preferred month for advertisers to release their advertisements, I recorded the release month of each advertisement. I then correlated these release months with significant LGBTQ+ events occurring in those months to ascertain whether such events influenced the number of advertisements released.

## 3. Advertisement characters based on their identity

This classification was undertaken to identify the most frequently depicted gender or sexual identity within the advertisements, allowing us to assess the extent of diversity in the representation of the LGBTQ+ community. For non-fictional advertisements featuring real LGBTQ + individuals, I identified their identity based on their real-life identity. In fictional advertisements, character identity was determined through self-declaration, romantic preferences, character interactions, voice-over descriptions, hints in the advertisement's storyline, campaign descriptions, or accompanying press briefings. Transgender characters were further classified based on whether a cisgender or transgender actor portrayed them, involving research into the actor's real-life identity through social media profiles, interviews, and news articles.

## 4. Advertisements based on their themes

The majority of advertisements revolved around two primary themes: love (emphasising that love transcends gender and welcomes all individuals regardless of their sexual orientation) and

human rights (focused on the right to lead a dignified life, with the exception of the choice of partners, which was incorporated into the love theme).

To determine the prevailing gender identity or sexual orientation within advertisements featuring these themes, I categorised these advertisements by character's gender identity and sexual orientation. This categorisation enabled me to count the number of love-themed advertisements with trans and gay, lesbian, or gay and lesbian characters and the number of rights-themed advertisements with trans and gay, lesbian, or gay and lesbian characters. Advertisements featuring trans and lesbian/gay characters or neither trans nor gay nor lesbian identities were listed separately.

## 5. Advertisements based on characters' age

To accurately reflect society, advertisements must showcase characters across a broad spectrum of age groups rather than exclusively focusing on a specific age range. To assess the diversity in character age representation, I categorised advertisements as featuring either young characters (aged 18–30) or older characters (over 30 years old). Character age was determined based on physical attributes associated with age.

Subsequently, advertisements featuring young characters were further subdivided into those featuring young gay, lesbian, or both gay and lesbian characters, as well as those featuring young transgender characters. Similarly, advertisements with older characters were categorised into those with older gay, lesbian, or both gay and lesbian characters and older transgender characters.

Advertisements that included a mix of young and **older** characters or characters with sexual orientations and gender identities other than gay, lesbian, or transgender were recorded separately.

## 6. Advertisements based on genre and LGBTQ+ character impact

At times, LGBTQ+ characters are portrayed in an exploitative manner solely to introduce a plot twist in a story. In order to investigate whether this holds true in the context of Indian LGBTQ+ advertisements, this study initially categorised advertisements into two groups: fictional (involving fictional characters within a fictional narrative) and non-fictional (featuring real individuals sharing authentic experiences). This categorisation aimed to discern which of these genres, if any, tend to utilise LGBTQ+ characters in an exploitative manner. Subsequently, each genre was scrutinised to ascertain whether the presence of LGBTQ+ characters significantly altered the storyline of the advertisement.

## 7. Advertisements based on their setting

The setting of an advertisement can reveal crucial details about a character's background, financial status, education, access to private and public spaces, and more. In some cases, advertisements may emphasise individuals from specific social strata over others. To address this question, we categorised advertisements as either urban or non-urban based on their settings.

'Urban settings were characterised by spacious homes, elegant interiors, upscale neighbourhoods, characters wearing branded clothing, fluent communication in polished English or Hinglish (a blend of Hindi and English), and the presence of expensive cars. In contrast, non-urban settings lacked these attributes, although it's important to note that non-urban settings do not exclusively refer to rural environments.

This methodology allowed me to comprehensively analyse Indian LGBTQ+ advertisement campaigns and draw meaningful conclusions.

## Results

### 1. Advertisement campaigns released annually

Between 2008, when the first LGBTQ+ campaign was observed, and 2021, when this research was conducted, Table 1 shows that a total of 76 companies launched 105 campaigns. With the exception of 2021, each subsequent year witnessed an increase in the number of advertisements released. Notably, the years 2018 and 2019 coincided with significant legal developments, including the decriminalisation of Section 377 (consequently legalising same-sex relationships) and the enactment of the Transgender Persons Act, ensuring the welfare and protection of transgender individuals [16]. These legal milestones were associated with a notable surge in advertisements targeting the LGBTQ+ community. However, the National Legal Services Authority vs. Union of India decision in 2014, recognised as "the first to legally recognise non-binary gender identities and uphold the fundamental rights of transgender persons in India" [17], did not result in considerable growth in the number of advertisements released that year.

### 2. Advertisements released monthly

Table 2 shows that there was a notable variation in the number of advertisements released each month. Specifically, the months of February, March, and June, corresponding to Valentine's Day, International Women's Day, and Pride Month, saw a substantial increase in the number of advertisements. In contrast, during the remaining months, the frequency of advertisements remained relatively low.

### 3. Characters based on their identity

Table 3 indicates that the advertisements showed that identities such as lesbians, gays, and trans women were more prominently represented compared to identities such as bisexual, trans men, and queer. Notably, lesbian, gay, and trans women (LBT) identities constituted a substantial majority, comprising 134 (76.14%) of the overall 176 characters. Consequently, for further analysis, only these three identity categories were considered.

Furthermore, among the 51 trans women characters featured in the advertisements, 6 (11.76%) were portrayed by cisgender actors, while 45 (88.24%) were portrayed by transgender

**Table 1. Advertisement campaigns released per year.**

| Year | Number of ads (in %) | Significant event of the year |
|---|---|---|
| 2008 | 1 (0.95%) | |
| 2010 | 2 (1.90%) | |
| 2011 | 1 (0.95%) | |
| 2013 | 2 (1.90%) | |
| 2014 | 2 (1.90%) | National Legal Services Authority v. Union of India |
| 2015 | 3 (2.86%) | |
| 2016 | 6 (5.71%) | |
| 2017 | 9 (8.57%) | |
| 2018 | 14 (13.33%) | Section 377 overturned |
| 2019 | 20 (19.05%) | Transgender Persons Act 2019 passed |
| 2020 | 24 (22.86%) | |
| 2021 | 21 (20.00%) | |
| Total | 105 | |

**Table 2. Advertisements released in each month.**

| Sr. no. | Campaign release month | No of ads (in %) | Days of significance in the month |
|---|---|---|---|
| 1. | January | 3 (2.86%) | |
| 2. | February | 14 (13.33%) | Valentine's Day |
| 3. | March | 12 (11.43%) | International Women's Day |
| 4. | April | 3 (2.86%) | |
| 5. | May | 6 (5.71%) | |
| 6. | June | 21 (20.00%) | Pride month |
| 7. | July | 5 (4.76%) | |
| 8. | August | 9 (8.57%) | |
| 9. | September | 9 (8.57%) | |
| 10. | October | 10 (9.52%) | |
| 11. | November | 9 (8.57%) | |
| 12. | December | 4 (3.81%) | |
| Total | - | 105 | - |

actors. Additionally, only trans men actors were cast in the roles of 2 trans men characters (Table 4).

## 4. Advertisements based on their themes

The analysis of advertisement campaigns revealed two predominant themes: love and human rights. Table 5 indicates that love-themed advertisements were featured in 43 (40.95%) instances, while human rights-themed advertisements were present in 38 (36.19%) instances.

In love-themed ads, 41 (39.05%) advertisements included gay, lesbian, or gay and lesbian characters, with an additional 2 (1.90%) advertisements featuring trans characters. Conversely, among the human rights-themed advertisements, 32 (30.48%) featured trans characters, and 6 (5.71%) included gay, lesbian, or gay and lesbian characters. Furthermore, 18 (17.14%) advertisements showcased both gay, lesbian, or gay and lesbian characters and trans characters, while 6 (5.71%) advertisements did not feature gay, lesbian, or trans characters.

## 5. Advertisements divided based on characters' age

Among the advertisements, as shown in Table 6, it was observed that 71 (67.62%) exclusively featured young characters, and 25 (23.80%) advertisements exclusively featured older

**Table 3. Characters classified based on their identity.**

| Sr. no. | Character's identity | Number of characters (in %) |
|---|---|---|
| 1. | Lesbian | 35 (19.89%) |
| 2. | Gay | 48 (27.27%) |
| 3. | Bisexual | 14 (7.95%) |
| 4. | Trans woman | 51 (28.98%) |
| 5. | Trans men | 2 (1.14%) |
| 6. | Nonbinary | 8 (4.55%) |
| 7. | Gender creative | 2 (1.14%) |
| 8. | Queer | 13 (7.39%) |
| 9. | Drag | 2 (1.14%) |
| 10 | Intersex | 1 (0.57%) |
| | Total | 176 |

**Table 4. Number of cisgender and transgender actors in the role of transgender persons.**

| Sr. no. | Number of trans woman characters | | Number of trans man characters | | Total |
|---|---|---|---|---|---|
| | Cisgender actors (in %) | Transgender actors (in %) | Cisgender actors (in %) | Transgender actors (in %) | |
| 1. | 6 (11.76%) | 45 (88.24%) | 0 (0%) | 2 (100%) | 6 |
| Total | 51 | | 2 | | 53 |

characters. The remaining 9 (8.57%) advertisements featured characters spanning all age groups, including children, young, and older individuals.

In advertisements featuring young characters, 47 (44.76%) showcased gay, lesbian, or gay and lesbian characters, whereas only 24 (22.86%) included trans characters. Conversely, in advertisements featuring older characters, 2 (1.90%) featured older gay, lesbian, or gay and lesbian characters, while 23 (21.90%) included older trans characters.

## 6. Advertisements classified by genre and impact of LGBTQ+ characters

Table 7 reports that 53 (50.48%) of the advertisements in this analysis were fictional in nature, and 52 (49.52%) were non-fictional.

Within the fictional advertisements, 21 (20.00%) featured LGBTQ+ characters that played a significant role in bringing about twists in the storyline. 32 (30.48%) had LGBTQ+ characters but did not introduce any significant plot twists.

In non-fictional advertisements, 1 (0.95%) advertisement integrated LGBTQ+ characters into the narrative to create a twist, and 51 (48.57%) advertisements with LGBTQ+ characters did not involve any major twists in the storyline.

## 7. Advertisements classified by setting

The advertisements were categorised based on their setting as follows: 58 (55.24%) advertisements were set in urban environments, and 24 (22.86%) advertisements were set in non-urban or other settings (Table 8).

In advertisements with an urban setting, 47 (44.76%) featured gay, lesbian, or gay and lesbian characters, and 11 (10.48%) included trans characters.

In non-urban settings, 2 (1.90%) advertisements had gay, lesbian, or gay and lesbian characters, and 22 (20.95%) advertisements featured trans characters.

23 (21.90%) advertisements were placed in a separate category as they included combinations of gay, lesbian, or gay and lesbian characters and transgender characters, or none of these identities.

## Discussion

### Low representation of LGBTQ+ characters in advertisements

The presence of LGBTQ+ characters in mainstream advertising emerged in the early 1990s in the United States [18, 19], whereas this study reveals that they appeared in Indian advertising

**Table 5. Advertisements based on their themes.**

| Sr. no. | Number of love-themed ads | | Number of rights-themed ads | | Ads with G, L, T, and L or T characters (in %) | Ads without G, L, or T characters (in %) | Total |
|---|---|---|---|---|---|---|---|
| | G, L, or G and L characters (in %) | T characters (in %) | T characters (in %) | G, L, and G or L characters (in %) | | | |
| 1. | 41 (39.05%) | 2 (1.90%) | 32 (30.48%) | 6 (5.71%) | 18 (17.14%) | 6 (5.71%) | 105 |
| Total | 43 (40.95%) | | 38 (36.19%) | | 18 (17.14%) | 6 (5.71%) | 105 |

**Table 6. Advertisements based on characters' age.**

| Sr. no. | Number of ads with young characters | | Number of ads with older characters | | Number of ads with mixed-aged characters (in %) | Total |
|---|---|---|---|---|---|---|
| | G, L, and G or L characters (in %) | T characters (in %) | G, L, and G or L characters (in %) | T characters (in %) | | |
| 1. | 47 (44.76%) | 24 (22.86%) | 2 (1.90%) | 23 (21.90%) | 9 (8.57%) | |
| Total | 71 (67.62%) | | 25 (23.80%) | | 9 (8.57%) | 105 |

later, in the late 2000s. This observation underscores that LGBTQ+ advertisements in India are still in their nascent stages compared to their counterparts in the United States. Remarkably, between 2008 (the year when LGBTQ+ advertising campaigns first appeared) and 2021 (the year of this research), a mere 105 such campaigns were produced. This number, despite India holding the ninth position globally in terms of market size and the fifth position in increased advertising expenditure in 2022 [20], indicates a hesitancy among businesses to invest in LGBTQ+ campaigns.

Despite the recent surge in LGBTQ+-centered campaigns in the United States [21, 22], the utilisation of gay images in targeted advertising remains relatively low compared to overall advertisement data [21]. In the United Kingdom, individuals with disabilities and LGBTQ + individuals are also underrepresented in comparison to the general population [23]. Globally, when contrasted with the 10% of individuals worldwide who identify as LGBTQ+, only a scant 1.8% of characters with an identifiable sexual orientation in advertisements identify as LGBTQ+ [24].

The primary factor contributing to this dearth of LGBTQ+ representation in advertisements is the fear of potential backlash from cisgender heterosexual consumers [25–28]. A survey reveals that 25% of Indian Millennials and 22% of Indian Gen Z, figures surpassing the global average, have boycotted businesses due to a misalignment of beliefs or actions [29]. Consequently, advertisers are often apprehensive about featuring LGBTQ+ characters in their ads due to risks associated with consumer backlash.

## Surge in advertisement numbers following the decriminalisation of Section 377 and the Transgender Person's Act

The remarkable increase in the number of LGBTQ+ advertisements in India from 2018 onward suggests that changes to the law encouraged companies to shed their hesitations and openly express their support for the LGBTQ+ community.

To address Davies' [1] query regarding whether commercial portrayals of non-heterosexual individuals have kept pace with the evolving social and legal landscape, I find that in India's case, they indeed have. Decriminalisation and the introduction of the Transgender Individuals Act align with the surge in LGBTQ+ commercials. Whitaker [30] (1999, p. 148) noted that LGBTQ+-centric marketing campaigns could only thrive if the gay and lesbian communities became more visible in everyday life and their cultures became apparent to the heterosexual majority. Prior to decriminalisation, companies were apprehensive about potential boycotts,

**Table 7. Advertisements classified by genre and impact of LGBTQ+ characters.**

| Sr. no. | Number of fictional ads | | Number of non-fictional ads | | Total |
|---|---|---|---|---|---|
| | Characters bring twists (in %) | Characters do not bring twists (in %) | Characters bring twists (in %) | Characters do not bring twists (in %) | |
| 1 | 21 (20.00%) | 32 (30.48%) | 1 (0.95%) | 51 (48.57%) | |
| Total | 53 (50.48%) | | 52 (49.52%) | | 105 |

**Table 8. Advertisements classified by setting.**

| Sr. no. | Number of ads with urban setting | | Number of ads with non-urban setting | | Others | Total |
|---|---|---|---|---|---|---|
| | G, L, or G and L characters | T characters | G, L, or G and L characters | T characters | | |
| | 47 (44.76%) | 11 (10.48%) | 2 (1.90%) | 22 (20.95%) | 23 (21.90%) | 105 |
| Total | 58 (55.24%) | | 24 (22.86%) | | | |

and LGBTQ+ individuals themselves feared persecution [31, 32]. To some extent, these concerns subsided following decriminalisation, motivating LGBTQ+ communities to advocate for their cause through advertisements.

Apart from the legal changes, another factor contributing to the rise in LGBTQ+ advertisements is the growing acceptance of LGBTQ+ communities among the general public, both in India and globally [33–36]. Nevertheless, many LGBTQ+ advocates argue that, instead of true acceptance and integration, gay individuals have sometimes been commodified and embraced only as the latest marketing trend [37]. Davies also raises doubts, questioning whether changes in perception (or merely depiction) are primarily driven by a broader social awareness among advertisers or by market-driven motives [1].

## Surge in advertisement campaigns during February, March, and June

The months of February, March, and June witnessed a significant upsurge in advertisement output, primarily driven by Valentine's Day, International Women's Day, and Pride Month, respectively.

Valentine's Day advertisements often carry the themes of "love is love" and "love without gender," promoting messages of inclusive love and relationships. Meanwhile, International Women's Day campaigns tend to emphasise the recognition of trans women as women, placing a strong focus on self-identification of gender rather than ascribing it as an inherent trait.

June experienced the most substantial spike in advertisement numbers, coinciding with the global celebration of Pride Month. This trend aligns with Western countries, where a substantial portion of LGBTQ+ campaigns are also launched in June [38, 39]. However, it is important to note that a potential shortcoming arises when companies engage with and represent LGBTQ+ individuals predominantly during Pride Month while remaining relatively invisible during other months of the year. To fully commit to the cause of inclusivity, companies should consider the need for year-round inclusive campaigns [39, 40].

## Lack of diversity in LGBTQ+ representation

Within Indian advertisements, gay, lesbian, and trans women identities emerge as the most prominent and visible. This dominance of gay and lesbian identities aligns with the historical trend where gender and sexuality discourse in the media often focuses on these identities. Numerous studies [18, 41–43] in the United States have previously pointed out the limited representation of transgender and bisexual identities in advertising. However, in the Indian context, trans woman identities are equally visible as gay and lesbian identities, while bisexual and trans man identities remain notably less represented.

One significant reason for the visibility of trans women identities in India can be traced back to the historical presence of the hijra identity—an indigenous South Asian identity that typically refers to individuals born male but identifying as feminine [44]. This identity has persisted in India for approximately three thousand years [45] and holds significant social visibility, which is reflected in various brands' media campaigns.

Despite the presence of diverse LGBTQ+ identities beyond gay, lesbian, and trans women, companies often fail to accurately represent the heterogeneity and authenticity of the LGBTQ+ population as a minority group [41]. Therefore, it is crucial to address this lack of diversity in LGBTQ+ representation in advertising.

A noteworthy observation in terms of identity representation is the relatively low number of cisgender actors playing trans characters. This phenomenon can be attributed to the prevalence of non-fictional genres in Indian advertisements, which frequently feature real individuals sharing their autobiographical narratives.

## Major themes: universal nature of love and human rights

In U.S. advertisements, the predominant themes revolve around the concepts of "love is love" and "all types of families are wholesome" [12]. However, in the context of India, I predominantly observe the former theme, as depictions of families with same-sex couples are notably absent. This difference can be attributed to the fact that India has not yet legalised same-sex marriage or civil unions.

An intriguing perspective emerging from this study is that advertisements featuring trans characters tend to revolve around the theme of human rights, while those with gay and lesbian characters primarily focus on love. This observation finds its roots in two significant legislative acts: the decriminalisation of Section 377 by the Supreme Court of India and the passage of the Transgender Persons (Protection of Rights) Act 2019 by the Indian Parliament.

Regarding Section 377, the media attention following decriminalisation primarily centred on gay sex, as reflected in the following headlines:

- "Section 377: Former Law Minister Kapil Sibal explains why UPA didn't bring in law decriminalising gay sex" [46].

- "Gay sex is legal in India, rules Supreme Court" [47].

- "India joins 25 nations where homosexuality is legal as SC decriminalises gay sex" [48].

- "India court legalises gay sex in landmark ruling" [49].

These headlines highlighted how the court's ruling allowed same-sex partners to engage in consensual sexual activity. However, they often overlooked that this law applied to any sexual activity deemed to be "against the order of nature," encompassing various acts beyond same-sex relations. Given that gay men were the population most affected by this colonial-era law, the media predominantly emphasised gay sex. Consequently, for advertisements featuring gay or lesbian characters, the theme of love remained dominant, as romantic love and sexual relationships are closely intertwined.

Turning to the Transgender Persons Act, discussions surrounding it, including media coverage and the act itself, predominantly focused on human rights issues, while overlooking the right to marry and live with a chosen partner. This focus is evident in the following headlines from articles covering the passage of the bill:

- "Transgender Bill Legitimizes Violence Against the Community" [50].

- "Organizations Protest Against Transgender Protection Bill for Treating Rape Against Trans-Women As 'Petty Crime'" [51].

- "Transgender rights bill makes way for self-declaration of identity, drops screening committee" [52].

- "Transgender Persons Bill, 2019: Dehumanising India's third gender with half-baked certification process" [53].

As a result, advertisements featuring trans characters often depict them in challenging environments, fighting for their rights and survival rather than portraying them in loving relationships.

## Disparities in portrayals: Poverty-stricken transgender individuals and affluent gay and lesbian individuals

In both fictional and non-fictional advertisements, advertisers often depict gay and lesbian individuals as young, financially stable, English-speaking, and residing in metropolitan areas in spacious homes. However, it is essential to note that this portrayal largely ignores the approximately 65% of Indians who reside in rural areas [54]. Various articles [55–58] have highlighted the unfortunate truth that LGBTQ+ individuals in rural India remain largely unheard and invisible in the media landscape.

When advertisements do delve into depictions of poverty and ageing, these aspects tend to be reserved for transgender women characters. Multiple studies have shed light on the substantial challenges faced by transgender women in India, including issues related to employment, healthcare, and social acceptance [59–64]. Due to their limited purchasing power, businesses often do not consider them as part of their target market. Instead, they utilise images of transgender individuals facing adversity to evoke sympathy from heterosexual audiences, thereby enhancing their brand image.

These trends are consistent with patterns of representation in other countries. The absence of diversity in terms of age and class is also observed in U.S. advertisements, primarily due to the absence of commercials targeted at children or older people [40]. Sender [65] argues that advertisers tend to focus on affluent, educated gay individuals residing in metropolitan areas because they possess substantial spending power. Consequently, their marketing efforts are geared towards this demographic, often overlooking lower-caste gay men living in remote areas, individuals with bisexual identities, or economically disadvantaged transgender individuals.

## Exploitative use of LGBTQ+ characters for plot twists

The use of LGBTQ+ characters in marketing has the potential to capture the public's attention, making it an enticing strategy for advertisers [66]. It is often seen as a means to add an element of novelty and excitement to mainstream consumption [4]. This approach is particularly evident in fictional advertising campaigns in India, where brands employ LGBTQ+ characters to introduce plot twists that can evoke various emotions such as surprise, shock, or amusement among the audience. These characters serve as a tool for advertisers to make their campaigns stand out in a crowded advertising landscape and resonate with heterosexual consumers.

However, it is essential to acknowledge that this exploitative use of LGBTQ+ images can have detrimental effects on the LGBTQ+ community instead of empowering it.

## Conclusion

This research paper draws several key conclusions:

1. LGBTQ+ representation in advertisements in India exhibited sluggish growth before 2018. However, a rapid surge in LGBTQ+-focused advertising followed decriminalisation and the enactment of the Transgender Persons (Protection of Rights) Act. This suggests that advertisers were cautious, waiting for a favourable legal and social landscape before embracing LGBTQ+ campaigns.

2. Despite the increase in the number of LGBTQ+ advertisements over the years, they still constitute a small fraction of marketing campaigns. There is a pressing need for a greater number of campaigns featuring LGBTQ+ characters.

3. The majority of LGBTQ+-focused campaigns were concentrated in June, February, and March, coinciding with specific events. However, a more balanced approach is needed, with campaigns running throughout the year rather than being confined to these select months.

4. While the advertisements predominantly featured gay, lesbian, and trans woman identities, other LGBTQ+ identities such as trans men, bisexual, queer, and intersex people remained largely invisible. Advertisers should strive for more inclusive representation, ensuring that no particular identity is highlighted at the expense of others.

5. Stereotypical portrayals were evident, with gay men and lesbians depicted as young, affluent, urban dwellers deeply in love, while trans women were often shown as older, marginalised, living in non-urban areas, and struggling for basic human rights. Brands must work towards portraying a diverse range of non-stereotypical characters in their advertisements.

6. The prevalence of non-fictional advertisements featuring real LGBTQ+ individuals sharing their authentic experiences was a commendable practice. However, there is a need to cast community members residing in non-metro areas to achieve more comprehensive representation.

7. Substantial numbers of advertisements still do not employ authentic portrayals of LGBTQ + individuals. Rather, they use LGBTQ+ individuals for shock value or to introduce unexpected story twists, reducing these individuals to stereotypes used either for amusement or to evoke pity from heterosexual viewers.

8. A positive aspect highlighted in this study is that a significant number of transgender characters were portrayed by transgender actors in advertisements.

The findings presented here reflect the current state of LGBTQ+ representation in Indian advertising, revealing inaccuracies and disparities in the portrayal of LGBTQ+ people. While it is recognised that companies primarily aim to expand their economic base and increase revenues [22], it is imperative that advertisers ensure their representations do not further alienate the LGBTQ+ community [12]. Advertising has a historical track record of influencing communities' perceptions and beliefs, particularly in the case of women [22]. As the most influential cultural medium of our times [41], advertising carries significant responsibility. This paper serves as a roadmap for rectifying past mistakes, and it is anticipated that the quality of future campaigns will contribute actively to the betterment of the LGBTQ+ community.

## Limitations and further research

This study has the following limitations:

1. A significant limitation arises from the necessity to assume a character's identity when it was not explicitly presented in the advertisement. The methodology section outlines the use of various factors to deduce identities. However, it is essential to emphasise that a person's self-declaration is the only accurate way to ascertain their gender identity or sexual orientation, and relying on assumptions based on external factors may not always be precise. In fictional advertisements, it was often impossible to discern a character's identity without resorting to such assumptions, hence the method employed in this study.

2. Another limitation is the potential subjectivity in perceiving whether an advertisement character is young or older or whether the setting is urban or non-urban. The researcher's own identities, including being a Hindu, single, Gujarati, Indian, 36 years old, cisgender gay man, belonging to a backward caste, raised in a lower-middle-class family, with access to higher education, among others, might have influenced the interpretation of the advertisements. Despite efforts to categorise aspects of advertising objectively, some degree of subjectivity may persist.

To build upon this study, future research can explore the following avenues:

1. While this study focuses on advertisements within mainstream media, an interesting area for future research would be to delve into LGBTQ+ niche media. Analysing advertisements specifically tailored for LGBTQ+ audiences could provide valuable insights into how these campaigns differ in terms of representation, themes, and strategies.

2. Another potential research direction is to explore LGBTQ+ print advertisements, distinct from audiovisual advertisements. This would involve a detailed analysis of print and static visual campaigns, which may offer unique perspectives on LGBTQ+ representation in advertising. Comparing these static representations with audiovisual ones could yield intriguing findings.

## Author Contributions

**Conceptualization:** Khuman Bhagirath Jetubhai.

**Data curation:** Khuman Bhagirath Jetubhai.

**Formal analysis:** Khuman Bhagirath Jetubhai.

**Funding acquisition:** Khuman Bhagirath Jetubhai.

**Investigation:** Khuman Bhagirath Jetubhai.

**Methodology:** Khuman Bhagirath Jetubhai.

**Project administration:** Khuman Bhagirath Jetubhai.

**Resources:** Khuman Bhagirath Jetubhai.

**Software:** Khuman Bhagirath Jetubhai.

**Supervision:** Khuman Bhagirath Jetubhai.

**Validation:** Khuman Bhagirath Jetubhai.

**Visualization:** Khuman Bhagirath Jetubhai.

**Writing – original draft:** Khuman Bhagirath Jetubhai.

**Writing – review & editing:** Khuman Bhagirath Jetubhai.

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
