## [Decision Letter · Decision Letter 0]

11 Sep 2023

PONE-D-23-17696LGBTQ+ Identities in the Indian Audiovisual Advertisements: A Content AnalysisPLOS ONE

Dear Dr. Khuman,

Thank you for submitting your manuscript to PLOS ONE. After careful consideration, we feel that it has merit but does not fully meet PLOS ONE’s publication criteria as it currently stands. Therefore, we invite you to submit a revised version of the manuscript that addresses the points raised during the review process.

The comments are mentioned at the end of the email.

We look forward to receiving your revised manuscript.

Kind regards,

Bidisha Banerjee, Ph.D.

Academic Editor

PLOS ONE

Journal Requirements:

Reviewers' comments:

Reviewer's Responses to Questions

**Comments to the Author**

1. Is the manuscript technically sound, and do the data support the conclusions?

Reviewer #1: Yes

Reviewer #2: Partly

2. Has the statistical analysis been performed appropriately and rigorously? 

Reviewer #1: Yes

Reviewer #2: N/A

3. Have the authors made all data underlying the findings in their manuscript fully available?

Reviewer #1: Yes

Reviewer #2: No

4. Is the manuscript presented in an intelligible fashion and written in standard English?

Reviewer #1: Yes

Reviewer #2: Yes

5. Review Comments to the Author

Reviewer #1: This is a good piece of work. I really enjoyed reading it. It adds to the literature. I have only one suggestion. The discussion, result, and conclusion look very repetitive, you may consider cutting the discussion down a little bit.

Reviewer #2: Hi,

First of all, many congratulations to the author for working on such an important topic. Although, one can notice author’s hard work here but this paper still lacks scientific rigor. I have a few following suggestions to improve the quality of this paper:

1. Comments:

2. URL of the data is not working

3. Take professional help for editing.

4. Under introduction, my suggestion to the authors would be add a few more studies linking advertisement with identity development among in-general population as well as among minority group of sexual preference.

5. What is LGBTQ+2 ? (Written in 3rd para of intro.)

6. No doubt this paper is based on an important aspect of our society that still needs to be studied by the Indian researchers. However, this paper lacks a strong rationale. It is my advice that the authors should write more convincing and stronger rationale for this study.

7. No need to write importance of this study under introduction section.

8. Under methodology, you have written that 16 web sources were traced for advertisement campaigns. Suggest you to give a reason for choosing these 16 web sources. On what basis these 16 web sources were shortlisted for this study?

9. The title suggest that the paper is based on a content analysis but except title of the paper there is no mention or description how the author have used this method for analysis.

10. Did you follow any criteria w.r.t. timeline of advertisement or language (as India is a country of many languages).

11. Your statement, “Thus, 105 observations of marketing campaigns were collected and used for the various analyses discussed below:”, is very unclear. What do you mean by ‘various analyses? As per my understanding, you are referring to various coding categories. Further, it is again unclear, whether these are inductive or deductive categories and on what basis you have decided on these so-called ‘various analyses?

12. Why did you analyses advertisements released yearly and monthly? You need to justify that too.

13. Further, you should have first developed research questions and hypothesis (you can find many good papers on content analysis that have developed hypothesis) to justify the requirement of your ‘various analyses’ mentioned under methodology.

14. Findings of this study is interesting but it lacks scientific rigour.

15. Suggest you to not use bullet points for writing your discussion.

16. Under 1st point of discussion, please correct the citation of (Group M, 2022).

6. PLOS authors have the option to publish the peer review history of their article (what does this mean?). If published, this will include your full peer review and any attached files.

Reviewer #1: No

Reviewer #2: No

---

## [Author Response · Author response to Decision Letter 0]

23 Sep 2023

Rebuttal Letter

2. URL of the data is not working. 

Response: All URLs listed in the Reference section have been verified, and any broken links have been replaced with working URLs.

3. Take professional help for editing. 

Response: The article has been edited and proofread in accordance with the suggestions provided by the reviewer.

4. Under introduction, my suggestion to the authors would be add a few more studies linking advertisement with identity development among in-general population as well as among minority group of sexual preference.

Response: This suggestion has been implemented, and additional studies discussing the link between advertisement and identity development have been included in the introduction section.

5. What is LGBTQ+2 ? (Written in 3rd para of intro.) 

Response: The superscript "2," which referred to an endnote number, has been merged with the main text since the journal guidelines do not permit separate food notes or endnotes.

6. No doubt this paper is based on an important aspect of our society that still needs to be studied by the Indian researchers. However, this paper lacks a strong rationale. It is my advice that the authors should write more convincing and stronger rationale for this study.

Response: In accordance with the reviewer's feedback, the rationale of the study has been included in the introduction section.

7. No need to write importance of this study under introduction section. 

Response: The guidelines from PLOS ONE recommend incorporating the significance of this study within the introduction.

8. Under methodology, you have written that 16 web sources were traced for advertisement campaigns. Suggest you to give a reason for choosing these 16 web sources. On what basis these 16 web sources were shortlisted for this study? 

Response: The justification for selecting these 16 web sources has been included in the methodology section.

9. The title suggest that the paper is based on a content analysis but except title of the paper there is no mention or description how the author have used this method for analysis.

Response: An explanation of the content analysis method has been included in the methodology section for clarity and comprehension. The same section contains seven bullet points that outline the application of content analysis in the paper.

10. Did you follow any criteria w.r.t. timeline of advertisement or language (as India is a country of many languages). 

Response: All advertisements, regardless of language and release time, have been selected, and these criteria are now explicitly stated in the methodology section.

11. Your statement, “Thus, 105 observations of marketing campaigns were collected and used for the various analyses discussed below:”, is very unclear. What do you mean by ‘various analyses? As per my understanding, you are referring to various coding categories. Further, it is again unclear, whether these are inductive or deductive categories and on what basis you have decided on these so-called ‘various analyses? 

Response: To eliminate any ambiguity, I have included an explanation of the fundamental content analysis process. The selection criteria for these coding categories have already been outlined in the methodology section.

12. Why did you analyses advertisements released yearly and monthly? You need to justify that too. 

Response: In the methodology section, we provide the rationale for categorising advertisement release data into yearly and monthly criteria. To repeat, this categorisation serves the purpose of identifying which specific month of the year sees the highest volume of advertisements released. Additionally, it helps us understand the reasons behind variations in advertisement releases, such as higher numbers during Pride Month or lower numbers during non-Pride months. Moreover, it allows us to investigate the increased number of advertisements released after 2018, which can be attributed to the decriminalisation of Section 377.

13. Further, you should have first developed research questions and hypothesis (you can find many good papers on content analysis that have developed hypothesis) to justify the requirement of your ‘various analyses’ mentioned under methodology.

Response: Research questions and hypothesis have been incorporated.

14. Findings of this study is interesting but it lacks scientific rigour. 

Response: In this study, content analysis is employed to quantify the occurrence of specific advertisement types or those featuring particular characters. The resulting counts are then interpreted through basic statistical techniques, such as correlation, percentages, and summation, using Microsoft Excel software. Notably, this paper does not employ advanced or intricate statistical methods to derive its findings. This may have led the reviewers to question its scientific rigour. Nevertheless, it's important to note that content analysis is a well-established research methodology, and like any other approach, it has its limitations, which are discussed in the research paper's limitations section.

15. Suggest you to not use bullet points for writing your discussion. 

Response: Bullet points have been replaced with subsection headings. 

16. Under 1st point of discussion, please correct the citation of (Group M, 2022). 

Response: The working URL has been inserted.

---

## [Decision Letter · Decision Letter 1]

25 Oct 2023

LGBTQ+ identities in the Indian audiovisual advertisements: A content analysis

PONE-D-23-17696R1

Dear Dr.Bhagirath,

We’re pleased to inform you that your manuscript has been judged scientifically suitable for publication and will be formally accepted for publication once it meets all outstanding technical requirements.

Kind regards,

Alex Siu Wing Chan

Academic Editor

PLOS ONE

Reviewers' comments:

Reviewer's Responses to Questions

**Comments to the Author**

1. If the authors have adequately addressed your comments raised in a previous round of review and you feel that this manuscript is now acceptable for publication, you may indicate that here to bypass the “Comments to the Author” section, enter your conflict of interest statement in the “Confidential to Editor” section, and submit your "Accept" recommendation.

Reviewer #2: All comments have been addressed

Reviewer #3: (No Response)

Reviewer #4: All comments have been addressed

2. Is the manuscript technically sound, and do the data support the conclusions?

Reviewer #2: Yes

Reviewer #3: Partly

Reviewer #4: Yes

3. Has the statistical analysis been performed appropriately and rigorously? 

Reviewer #2: N/A

Reviewer #3: Yes

Reviewer #4: Yes

4. Have the authors made all data underlying the findings in their manuscript fully available?

Reviewer #2: Yes

Reviewer #3: Yes

Reviewer #4: Yes

5. Is the manuscript presented in an intelligible fashion and written in standard English?

Reviewer #2: Yes

Reviewer #3: Yes

Reviewer #4: Yes

6. Review Comments to the Author

Reviewer #2: Hi,

Thank you for addressing all my concerns and suggestions. However, I still have one suggestion to make.

Please do not write conclusion, limitations and future research using bullet points. It do not look academic or professional. Therefore, suggest you to remove it.

Other than this, I have no concerns.

Thank you.

Reviewer #3: This paper conducts a content analysis on advertisements to examine LGBTQ+ representation in marketing campaigns in India. The authors find that advertisements that depict sexual minorities centered around themes of love, with young actors in urban settings being overutilized. Advertisements using transgender adults centered around human rights with older actors in non-urban settings.

There are many strengths to this paper:

1) First, the authors make a strong “case” as to the contribution of their study. The introduction of the manuscript does a great job contextualizing their study within the broader work on LGBTQ advertising analysis while also making a great argument for the novel contribution that their analysis provides.

2) I appreciated the inclusion of notable cultural and legal events to speculate about the motivations for the increase in LGBTQ+ centered advertising.

3) Including an analysis of gender minorities in addition to sexual minorities really enhances the study. I liked the depth and complexity of considering gender and sexuality.

4) I think that overall, the findings are interesting and speak to broader cultural shifts in India regarding perceptions and attitudes towards gender and sexual minority people. This study is valuable and informative.

I highlight some concerns below:

1) I think that 7 RQ questions may be a bit too much. It takes away from a broader discussion, when the RQs are so pointed. Would it be possible to condense the RQs to 3? I think that many of the RQs are actually “sub questions.” I suggest that the authors reframe their questions into 3 broader questions that encompass some of these “sub questions.” For example, 1) How are gender and sexual minorities portrayed in advertisements? is a broader RQ that encompasses questions 3,4, 5 and 7.

2) This is minor, but there seems to be variation in text font size throughout the manuscript.

3) While the author does a great job of describing the criteria for analysis, I am unsure about some additional elements of the observations. The author mentions that they were examining audio visual observations but fails to mention the length of the observations. For example, did the audio-visual observations range in length? If so, how did they address differences in length? Often, people conducting content analysis will select a specific time frame to examine such as 30 seconds to maintain consistency across observations. More clarification would be helpful.

4) It seems that the author was looking for “love” and “human rights” themes since this was posed as a RQ. Was there some reasoning for this? Were there other themes that arose or were the authors only interested in the two selected themes.

5) I would avoid using bullet points in the discussion of the results (pages 29-30).

6) There is a significant number of subheadings throughout the text. It “breaks” the narrative and at times is a bit distracting. I would suggest reorganizing the manuscript in a way that relies on transition sentences, rather than on subheadings.

7) Numbering the conclusions is not the best way to discuss the results in the conclusion. I would present this in narrative form. Discuss the findings, rather than list them.

8) Again, the limitations or future implications should not be simply listed. Re-write in narrative form and present more of a discussion.

9) I understand that the analysis was more descriptive without statistical modeling. I do not think that is a weakness. I would, however, encourage the authors to make a “bigger picture” conclusion. Moving beyond listing the results but speaking to the overarching literature on sexuality and marginalization would strengthen the paper.

Reviewer #4: It appears the author has addressed all of the comments from previous reviews. I have nothing additional to add.

7. PLOS authors have the option to publish the peer review history of their article (what does this mean?). If published, this will include your full peer review and any attached files.

Reviewer #2: No

Reviewer #3: No

Reviewer #4: No

---

## [Editor Report · Acceptance letter]

31 Oct 2023

PONE-D-23-17696R1 

LGBTQ+ identities in the Indian audiovisual advertisements: A content analysis 

Dear Dr. Khuman:

I'm pleased to inform you that your manuscript has been deemed suitable for publication in PLOS ONE. Congratulations! Your manuscript is now with our production department. 

Kind regards, 

on behalf of

Dr. Alex Siu Wing Chan 

Academic Editor

PLOS ONE